# Route of oxytocin administration for preventing blood loss at caesarean section: a systematic review with meta-analysis

Maria Regina Torloni [1,2] Monica Siaulys,[3] Rachel Riera,[2,4]
Ana Luiza Cabrera Martimbianco,[2,5] Rafael Leite Pacheco,[2,6]
Carolina de Oliveira Cruz Latorraca [2] Mariana Widmer,[7] Ana Pilar Betran[7]

For numbered affiliations see end of article.

**Correspondence to**
Dr Maria Regina Torloni;
ginecologia@terra.com.br

## ABSTRACT

**Objectives** Assess the effects of different routes of prophylactic oxytocin administration for preventing blood loss at caesarean section (CS).

**Design** Systematic review and meta-analysis.

**Methods** Medline, EMBASE, CINAHL, Cochrane Library, BVS, SciELO and Global Index Medicus were searched through 24 May 2020 for randomised controlled trials (RCTs) comparing different routes of prophylactic oxytocin administration during CS. Study selection, data extraction and quality assessment were conducted by two investigators independently. We pooled results in fixed effects meta-analyses and calculated average risk ratio (RR), mean difference (MD) and 95% CI. We used GRADE to assess the overall quality of evidence for each outcome.

**Results** Three trials (180 women) were included in the review. All studies compared intramyometrial (IMY) versus intravenous oxytocin in women having prelabour CS. IMY compared with intravenous oxytocin administration may result in little or no difference in the incidence of postpartum haemorrhage (RR 0.14, 95% CI 0.01 to 2.70; N=100 participants; 1 RCT), hypotension (RR 1.00, 95% CI 0.29 to 3.45; N=40; 1 RCT), headache (RR 3.00, 95% CI 0.13 to 69.52; N=40; 1 RCT) or facial flushing (RR 0.50, 95% CI 0.05 to 5.08; N=40; 1 RCT); IMY oxytocin may reduce nausea/vomiting (RR 0.13, 95% CI 0.02 to 0.69; N=140; 2 RCTs). We are very uncertain about the effect IMY versus intravenous oxytocin on the need for additional uterotonics (RR 0.82; 95% CI 0.25 to 2.69; N=140; 2 RCTs). IMY oxytocin may reduce blood loss slightly (MD −57.40 mL, 95% CI −101.71 to −13.09; N=40; 1 RCT).

**Conclusions** There is limited, low to very low certainty evidence on the effects of IMY versus intravenous oxytocin at CS for preventing blood loss. The evidence is insufficient to support choosing one route over another. More trials, including studies that assess intramuscular oxytocin administration, are needed on this relevant question.

**PROSPERO registration number** CRD42020186797.

## INTRODUCTION

Deliveries by caesarean section (CS) have increased worldwide over the last three decades, including in low-income and middle-income countries (LMICs).[1] Even in optimal

### Strengths and limitations of this study

► This is the first systematic review to assess the effects of different routes of oxytocin administration for preventing bleeding in women giving birth by caesarean.

► We performed an extensive search without language restrictions in 10 databases to identify relevant trials, and followed strict methodology throughout the systematic review process.

► We found few trials, involving a small number of participants, and addressing a limited number of outcomes.

► The trials did not test all possible routes of oxytocin administration, and only included prelabour caesareans at term.

► The overall certainty of the evidence was low or very low for all outcomes due mainly to imprecision.

conditions, women who have a CS lose more blood, and have a higher risk for postpartum haemorrhage (PPH) than those who give birth by vaginal delivery (VD).[2 3] PPH is an important cause of maternal morbidity, including prolonged hospital stay, blood transfusion and hysterectomy, and it is the leading cause of maternal mortality worldwide.[4–6] Uterine atony is responsible for 50%–80% of all cases of PPH.[4 6]

Routine administration of prophylactic uterotonics soon after birth reduces the incidence of PPH and its associated complications.[6] Oxytocin remains the first-line agent for preventing PPH recommended by WHO because of its lack of major contraindications, similar efficacy and lower cost and incidence of side effects, compared with other uterotonics.[6] For women having a VD, a systematic review in 2020 estimated that intravenous oxytocin is more effective than intramuscular (IM) administration to reduce PPH, severe

PPH, blood transfusion and severe maternal morbidity, with no clear differences in adverse events.[7] Based on these findings, WHO updated its guideline in 2020 and now recommends that women giving birth by VD who already have endovenous access should receive a slow intravenous injection of 10 IU of oxytocin, in preference to IM administration.[8] For women giving birth by CS, due to the lack of specific evidence, WHO currently recommends 10 IU of oxytocin without preference for the intravenous or IM routes.[8] Other guidelines provide little, unclear or no recommendations on the route of administration of prophylactic oxytocin at CS.[9–11]

Since all women having a CS will have an endovenous access, intravenous administration of oxytocin seems logical. However, due to concerns about the cardiovascular side effects of the drug in women undergoing a major abdominal surgery, many studies have investigated different dosages and regimens of intravenous oxytocin administration at CS in search of the optimal method to safely prescribe this uterotonic.[12–17] After intravenous injection in late pregnancy, oxytocin has an almost immediate onset of action but a half-life of only 5–12 min due to its rapid hepatic and renal degradation.[17–19] With continued infusion, oxytocin reaches a steady state plateau in the plasma in approximately 40 min.[17–19] Activation of oxytocin receptors in the myometrium stimulates prostaglandin synthesis and increases intracellular calcium in myocytes which leads to strong uterine contractions thus reducing blood loss from the placental site.[16–19] However, the simultaneous activation of oxytocin receptors in the cardiovascular system causes decreased systemic vascular resistance, hypotension, tachycardia and coronary vasoconstriction with the attendant symptoms of skin flushing, headache, nausea, vomiting and chest pain.[17 20 21] These changes are usually transient, depend on the dose and speed of intravenous oxytocin administration, and are tolerated by most healthy young women.[16 22] However, in some women undergoing a CS, these haemodynamic changes can be potentially dangerous and even life-threatening because of the sympathetic nervous blockade secondary to regional anaesthesia, the concomitant use of other vasoactive drugs, excessive blood loss and the underlying clinical and obstetric disorders that may have led to the CS indication (eg, cardiac disease, hypertensive disorders, placenta previa).[21 23–25] Due to its antidiuretic properties, oxytocin administration can also lead to water intoxication resulting in pulmonary oedema, seizures, coma and even death, especially in women who receive large volumes of fluid.[13 17–19] The routine use of large intravenous fluid preloads during CS under regional anaesthesia can potentially increase the risk of this complication.[26]

After IM oxytocin injection, uterine response occurs within 3–5 min and can last for up to 2–3 hours.[8 18 19 27] Direct intramyometrial (IMY) injection of uterotonics can also promote uterine contraction.[28 29] IMY oxytocin injection is believed to promote immediate uterine contraction by local effect and by drug absorption from the myometrium into the systemic circulation.[28 30–32] Although oxytocin is not licensed for IMY use,[18 19] this route of administration has been tested for preventing PPH at CS since 1990.[28 30–32]

The route of administration of prophylactic oxytocin at CS can potentially affect the volume of maternal blood loss as well as the incidence of drug-related adverse effects. Since the side effects of oxytocin are directly related to dose and rate of administration,[13 17–19 27] in theory, the IM or IMY routes could lead to fewer cardiovascular side effects than the intravenous route. If the efficacy of IM and IMY prophylactic oxytocin during CS is similar to intravenous administration, and if these routes have fewer side effects than the endovenous route, this could challenge the preference for intravenous administration. Several randomised trials have compared different routes of prophylactic oxytocin administration at CS, but there are no systematic reviews on this topic. It is important to assess the balance between the effectiveness and adverse effects of the various routes of prophylactic oxytocin administration at CS because this information is essential to underpin evidence-based recommendations on the prevention of PPH in women given birth by CS.

## OBJECTIVES

To assess the effects of different routes of prophylactic oxytocin administration for preventing blood loss at CS.

## METHODS

We conducted this review following Cochrane methods[33] and reported it according to the Preferred Reporting Items for Systematic Reviews and Meta-Analyses (PRISMA) statement.[34] We registered the protocol prospectively (CRD42020186797-online supplemental file 1).

### Patient and public involvement

There was no patient or public involvement in this study.

### Types of studies

We included randomised clinical trials with a parallel design. Abstracts were eligible if they provided sufficient outcome data and methodological details for quality assessment.

### Types of participants

Study participants were women having a primary or repeat CS, before or during labour, at any gestational age and for any indication, regardless of previous exposure to oxytocin (for labour induction or augmentation). Studies with women at high or low baseline risk for PPH, with or without comorbidities, and with singleton or multiple pregnancies were eligible for inclusion. We included studies involving women with both routes of delivery only if authors provided separate data for those giving birth by CS.

## Types of interventions

We included studies that compared different routes of oxytocin administration (intravenous, IM or IMY) given alone, at any moment during CS (predelivery, after delivery or after placental extraction). We excluded studies that administered oxytocin associated with other drugs for preventing PPH.

## Outcomes

Studies that reported at least one of our outcomes of interest were included. Our outcomes were based on the PPH prevention core outcome set created by the CROWN initiative.[35] The primary outcomes were PPH >1000 mL (measured objectively or subjectively), use of additional uterotonics and any immediate adverse effects possibly related to oxytocin (eg, nausea/vomiting, facial flushing, headache, hypotension, cardiac arrhythmia). The secondary outcomes were total volume of blood loss at CS, need for transfusion, maternal transfer to higher level of care, shock, severe maternal morbidity (admission to ICU, hysterectomy, coma or organ failure), maternal mortality related to PPH, breastfeeding at hospital discharge and maternal satisfaction.

## Search strategy

We created a search strategy with the following general terms and synonyms: "caesarean section" or "C-section" or "abdominal delivery" and "oxytocin" (details in online supplemental file 2). The search strategy was developed by an experienced information specialist. It was adapted and run in seven electronic databases (MEDLINE, EMBASE, Cochrane Library, CINAHL, BVS, Global Index Medicus, SciELO) and two trial registry platforms (ClinicalTrials.gov and WHO-ICTRP) (online supplemental file 2). We searched for unpublished studies (grey literature) in Opengrey (https://opengrey.eu). We screened the reference lists of all retrieved studies and relevant systematic reviews to identify additional potentially eligible trials not captured by the electronic searches. There were no language or publication status restrictions. Searches were run from database inception until 24 May 2020.

## Process of study selection, data extraction and risk of bias assessment

All references were uploaded in the Rayyan platform[36] and duplicates were excluded. All titles and abstracts were screened independently by two review authors. Potentially eligible studies were selected for full-text reading; those that fulfilled the selection criteria were included in the review. Two review authors extracted data independently and a third author check data for accuracy. Two review authors independently assessed the quality of included studies using the Cochrane Risk of Bias tool.[33] In each study, authors assessed and graded seven domains as being at high, low, or unclear risk of bias. Four domains (random sequence generation, allocation concealment, selective reporting and other source of bias) were assessed at study level; three domains (blinding of participants and personnel, blinding of outcome assessment, and incomplete outcome data) were assessed at outcome level. Any disagreements in the process of study selection, data extraction or risk of bias assessment were solved by a third senior review author.

## Data analyses and assessment of the certainty of the evidence

We planned to conduct three comparisons between routes of oxytocin administration: intravenous versus IM, intravenous versus IMY, and IM versus IMY. All analyses were conducted on an intention-to-treat basis. When data were missing, we contacted trial authors. Where meta-analyses were not deemed suitable, we report results descriptively. Similar study data were pooled using the software Review Manager V.5.4 (The Cochrane Collaboration, 2020). For dichotomous outcomes, we calculated risk ratio (RR) and 95% CI; for continuous data, we calculated mean differences (MD) and 95% CI. We pooled data using the Mantel-Haenszel fixed-effect method; $I^2 \geq 50\%$ was considered an indication of high statistical heterogeneity. We planned to conduct prespecified subgroup analyses for the primary outcomes (when data were available) according to parity (nullipara vs multipara and multipara with CS vs multipara without previous CS), type of CS (prelabour vs intrapartum), previous use of oxytocin in labour, and baseline risk for PPH. We could not conduct these analyses because of the small number of trials included and lack of information. We planned to conduct sensitivity analyses for the main comparisons restricted only to high-quality studies (at low risk of bias for random sequence generation and allocation concealment), and restricted only to studies that assessed blood loss using an objective measure. These analyses were not done because all studies measured blood loss objectively and none were high quality. Due to the small number of trials, we could not assess publication bias.

We assessed the certainty (quality) of the body of evidence for selected outcomes using the Grades of Recommendations, Assessment, Development and Evaluation (GRADE) approach.[37] Evidence was categorised as being of high, moderate, low, or very low quality. We downgraded the quality of the evidence one or two levels because of trial limitations, inconsistency, indirectness, imprecision, and publication bias.[37] This process was conducted by two independent investigators; disagreements were solved by a third senior investigator.

## RESULTS

A total of 13 389 unique references were identified. After title and abstract screening, six citations were selected for full-text reading. Two studies and one abstract were excluded (online supplemental file 3) and three trials[30–32] were included in the review (figure 1).

The three trials included a total of 180 women and were conducted between 1998 and 2012, in Japan,[30] Canada[31] and India[32] (table 1 and online supplemental file 4). None of the trials provided clear information on the

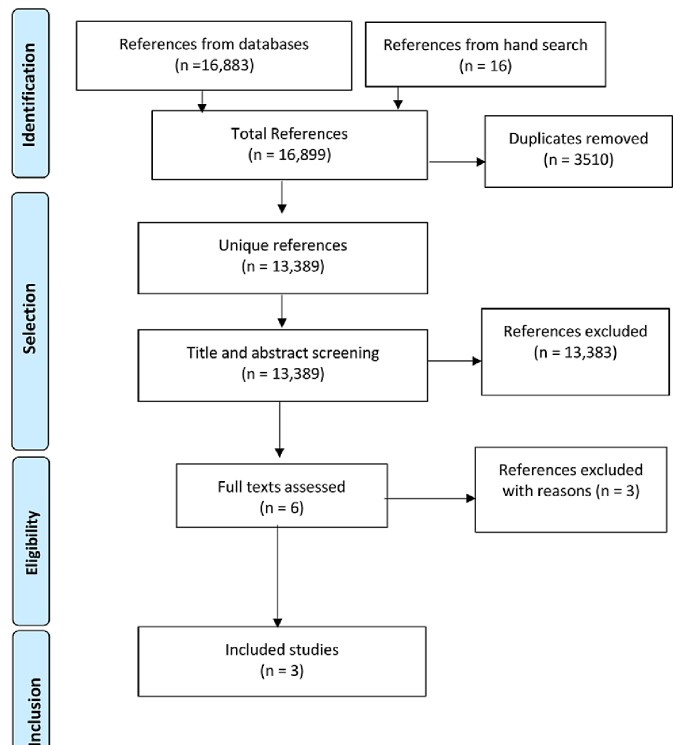

**Figure 1** Flow chart of the process of study selection.

proportion of nulliparas or multiparas (with and without previous CS). One trial[32] did not report participants' baseline risk for PPH, gestational age or type of CS; the other two trials[30 31] included participants with regular/low baseline risk for PPH, undergoing a prelabour CS with at least 36 weeks' gestation, under spinal anaesthesia (table 1). All three trials compared intravenous versus IMY oxytocin administration after fetal delivery. There was heterogeneity between studies on the oxytocin doses used in each route; the IMY dose ranged from 4.3 IU[30] to 20 IU,[31] and the total intravenous dose ranged from 4.3 IU[30] to 20 IU.[32] One trial[30] administered oxytocin with the placenta still in place, immediately after umbilical cord clamping, while the other two studies waited for placental detachment or removal before giving the oxytocin.[31 32] Akinaga et al randomised 40 Japanese women to receive either an IMY oxytocin injection and an intravenous saline bolus injection, or an intravenous oxytocin bolus and an IMY saline injection immediately after umbilical cord clamping; all women also received a slow oxytocin maintenance infusion for the next 5 hours.[30] Dennehy et al randomised 40 women in Canada to receive an IMY oxytocin bolus injection into the uterine fundus and a saline intravenous bolus injection, or an intravenous oxytocin bolus injection and a saline IMY injection after placental removal.[31] Mangla et al randomised 150 women in India to start a 500 mL infusion containing oxytocin after placental separation (group 1, N=50), or to receive an IMY oxytocin bolus injection (half in each uterine cornus) either after placental separation (group 2, N=50) or before placental separation (group 3, N=50).[32] We did not include in this systematic review the participants in the third group

because it compared a different route and a different timing of oxytocin administration (online supplemental file 4). The two trials[30 32] that reported the volume of blood loss[30 32] used objective methods of assessment, but only one[32] assessed the incidence of blood loss >900 mL. All trials assessed the need for additional uterotonics and at least one adverse effect of oxytocin. No trial reported any of our other secondary outcomes (table 1 and online supplemental file 5).

Figure 2 presents the risk of bias of the three included studies (see online supplemental file 6) for details and summary graph). All studies had at least one domain with unclear or high risk of bias. Akinaga et al[30] and Mangla et al[32] did not describe the method used to generate the randomisation sequence; Dennehy et al[31] and Mangla et al[32] did not describe how allocation concealment was achieved. Overall, the three trials had an unclear risk of bias for random sequence generation or allocation concealment, or both domains. We contacted the authors of the three trials to obtain additional details, but none replied.

Table 2 presents a summary of all outcomes for the comparison between prophylactic IMY versus intravenous oxytocin in women having a CS. (See online supplemental file 7) for the assessment of the certainty of the evidence (GRADE) for each outcome.

### Incidence of PPH
None of the studies assessed PPH according to the standard definition (≥1000 mL). However, Mangla et al[32] reported that none of the women who received IMY oxytocin had a blood loss >900 mL compared with 6% of those who received intravenous oxytocin (0/50 vs 3/50, RR 0.14, 95% CI 0.01 to 2.70; 100 participants; 1 RCT, low certainty evidence) (table 2).

### Need for additional uterotonics
The pooled estimate from the three trials was very imprecise and compatible with an important risk reduction or increase in the need for additional uterotonic with IMY compared with intravenous oxytocin injection (4/70 vs 5/70, respectively; RR 0.82; 95% CI 0.25 to 2.69; 140 participants; 2 RCTs; $I^2$=61%, very low certainty evidence) (figure 3, table 2).

### Adverse effects of oxytocin
Hypotension: Data from one trial[30] indicate that IMY compared with intravenous oxytocin administration may have no effect on the incidence of isolated hypotension episodes that do not require ephedrine (RR: 1.00, 95% CI 0.29 to 3.45; 40 participants; 1 RCT; low certainty evidence). Two trials[30 31] (79 participants) assessed the incidence of persistent hypotension requiring treatment with epinephrine, but there were no events among the women who received IMY or intravenous oxytocin. Although the lack of events did not allow estimation of the RR, the evidence from these trials is consistent with

**Table 1** Main characteristics of trials on routes of oxytocin administration at CS

| Characteristics | Akinaga et al[30] | Dennehy et al[31] | Mangla et al[32] |
|---|---|---|---|
| Setting | Japan, one university hospital | Canada, no information on type of hospital(s) | India, one hospital |
| Design | Double blind, placebo controlled RCT | Double blind, placebo controlled RCT | RCT |
| Sample size | 40 | 40 | 100* |
| Gestational age | All ≥36 weeks | All ≥37 weeks | No information |
| Baseline risk for PPH | All low risk | All low risk | No information |
| Parity | Unclear | No information | No information |
| Participants with previous CS | Unclear | No information | No information |
| Type of CS | Prelabour, scheduled | Prelabour, scheduled | No information |
| Previous exposure to oxytocin | No | No | No information |
| Anaesthesia | Spinal | Spinal | Spinal or general |
| Timing of administration | After umbilical cord clamping | After placental removal | After placental separation |
| Oxytocin routes and regimen | IMY bolus versus intravenous bolus followed by intravenous maintenance infusion | IMY bolus versus intravenous bolus | IMY bolus versus intravenous infusion |
| Oxytocin dose, diluent, speed of administration, (dose/min), Total dose | IMY bolus: 4.3 IU† in 2 mL 0.9% saline over 30 seconds in uterine fundus. Total dose: 4.3 IU Intravenous bolus: 4.3 IU† in 10 mL 0.9% saline over 30 seconds (8.5 IU/min) Total dose: 4.3 IU Maintenance infusion (both groups): 6.1 IU‡ in 500 mL Ringer at 100 mL/hour for 5 hours (0.02 IU/min) Total dose: 6.1 IU Total oxytocin regimen dose§: 10.4 IU | IMY bolus: 2 mL of 10 IU/mL in uterine fundus. Total dose: 20 IU Intravenous bolus: 0.5 mL of 10 IU/mL over unknown time Total dose: 5 IU | IMY bolus: 5 IU in 10 mL 0.9% saline (5 mL in each uterine cornu). Total dose: 5 IU Intravenous infusion 20 IU in 500 mL Ringer, no information on speed or duration Total dose: 20 IU |
| Outcomes reported | Volume of blood loss, additional uterotonics, adverse effects | Additional uterotonics, adverse effects | Volume of blood loss, additional uterotonics, adverse effects |

*Mangla 2012 had a third group with 50 women (total sample: 150) that was not included in this review because it administered IMY before placental separation (ie, different timing and route from intravenous group).
†Bolus dose was calculated based on the mean weight of participants (0.07 IU/kg × 61 kg).
‡Maintenance infusion dose was calculated based on the mean weight of participants (0.01 IU/kg × 61 kg).
§Total oxytocin dose received by all participants (bolus + maintenance infusion).
CS, caesarean section; IMY, intramyometrial; PPH, postpartum haemorrhage; RCT, randomised controlled trial.;

absence of a difference between IMY and intravenous oxytocin for this outcome. (table 2).

Changes in heart rate (HR): We could not perform a meta-analysis for this outcome. Akinaga et al[30] reported no apparent changes in the HR of the 20 women who received IMY oxytocin while the HR of the intravenous group started to increase soon after the bolus injection, reaching a maximum value 75 seconds after. The maximum HR in the intravenous group was significantly higher than in the IMY group (92.9±10.3 vs 83.2±7.2 beats per minute, MD: 9.7, 95% CI 4.0 to 15.4, p=0.002). On the other hand, Dennehy et al reported a significantly higher increase in HR 1 min after IMY than after intravenous oxytocin administration, but the authors did not provide numerical data for this outcome.[31] It should be noted that while Akinaga et al[30] gave the same dose of oxytocin (4.3 IU) in both routes, the IMY oxytocin dose given by Dennehy et al[31] was four times higher than the intravenous dose (20 IU vs 5 IU, respectively).

Nausea and/or vomiting: The pooled estimate of two trials including 140 women showed a lower average risk of nausea and/or vomiting in the group that received IMY oxytocin (RR 0.14; 95% CI 0.03 to 0.78; 140 participants; 2 RCTs; I²=0%, low certainty evidence) (figure 4, table 2).

Headache and facial flushing: Only Akinaga et al[30] reported the incidence of headache and facial flushing. One woman in the IMY group complained of headache versus none in the intravenous group (1/20 vs 0/20, RR 3.00, 95% CI 0.13 to 69.52; 40 participants; 1 RCT; low certainty evidence). One participant in the IMY and two in the intravenous oxytocin group had facial flushing (1/20 vs 2/20, RR 0.50, 95% CI 0.05 to 5.08; 40 participants; 1 RCT; low certainty evidence) (table 2).

## Volume of blood loss

Mangla et al[32] reported that the women in the IMY group had a lower mean volume of blood loss than those in the intravenous group (460 mL vs 606 mL, respectively), but

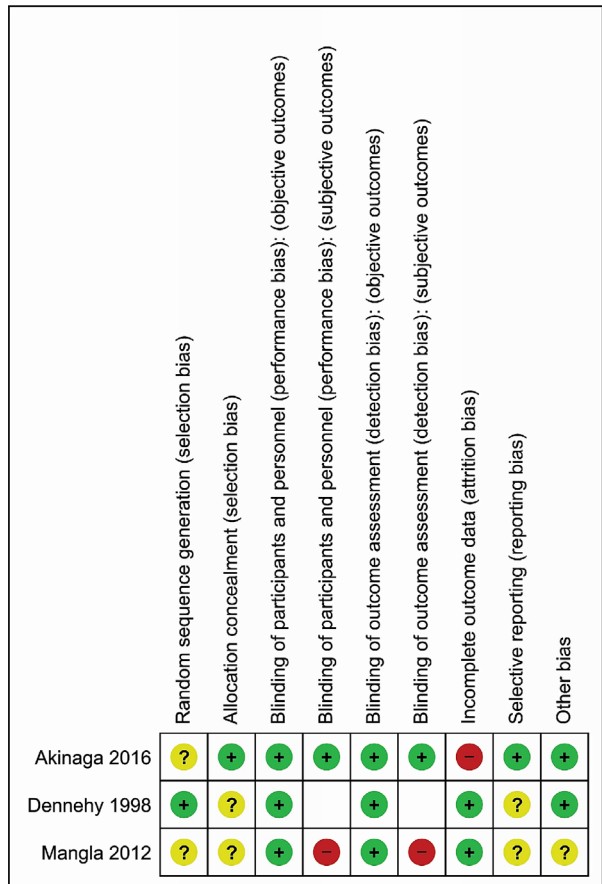

**Figure 2** Risk of bias summary of trials on oxytocin route of delivery at caesarean.

did not provide the SD or CIs for this measure. Akinaga *et al*[30] also reported a slightly lower mean volume of blood loss in the IMY than in the intravenous group (606.8±68.8 vs 664.2±74.1 mL; MD −57.40 mL; 95% CI −101.71 to −13.09; 40 participants; 1 RCT; low certainty evidence)

(table 2). We could not pool the results of the two trials because one of them[32] did not provide information on SD and the authors did not reply to our emails.

## DISCUSSION
### Main findings and interpretation

Our extensive search identified only three trials on the effects of different routes of administration of prophylactic oxytocin in women giving birth by CS. The existing evidence suggests that IMY compared with intravenous oxytocin administration may result in little to no difference on the incidence of PPH (≥1000 mL) and most side effects (hypotension, headache or facial flushing), but IMY oxytocin may reduce the incidence of nausea/vomiting. The evidence also suggests that IMY oxytocin may reduce the volume of blood loss slightly. We are very uncertain about the effects of IMY versus intravenous oxytocin on the need for additional uterotonics. The overall certainty of the evidence was low or very low for all outcomes due mainly to imprecision. According to the GRADE approach, this means that we have limited confidence in the effect estimates and that the true effect may be substantially different from the estimates presented.[37]

Where information was available, most trials recruited only healthy women with a low baseline risk for PPH, who were having an elective prelabour CS under spinal anaesthesia to deliver a single fetus at term. Information about parity, including the percentage of participants with one or more previous CS, was unclear or missing in all studies. Therefore, we cannot infer that the results of this systematic review apply to women with other characteristics, such as preterm or multiple pregnancies, women having intrapartum CS, or those with previous exposure to oxytocin for labour induction or augmentation. The different doses and regimen of oxytocin used in the trials

**Table 2** Comparison of prophylactic intramyometrial versus intravenous oxytocin for preventing blood loss in women giving birth by caesarean

| Outcome or subgroup title | N studies | N women | Effect measure (IMY vs intravenous) | Effect size (95% CI) | Certainty of the evidence (GRADE)* |
|---|---|---|---|---|---|
| 1. PPH >1000 mL | 1 | 100 | Risk ratio | 0.14 (0.01 to 2.70) | Low |
| 2. Use of additional uterotonics | 2 | 140 | Risk ratio | 0.82 (0.25 to 2.69)† | Very low |
| 3. Hypotension not requiring ephedrine | 1 | 40 | Risk ratio | 1.00 (0.29 to 3.45) | Low |
| 4. Hypotension requiring ephedrine | 2 | 79 | Risk ratio | Not estimable | Low |
| 5. Nausea and/or vomiting | 2 | 140 | Risk ratio | 0.13 (0.02 to 0.69)† | Low |
| 6. Headache | 1 | 40 | Risk ratio | 3.00 (0.13 to 69.52) | Low |
| 7. Facial flushing | 1 | 40 | Risk ratio | 0.50 (0.05 to 5.08) | Low |
| 8. Mean blood loss (mL) | 1‡ | 40 | Mean difference | −57.40 (−101.71 to -13.09) | Low |

*See online supplemental file 7 for details.
†Mantel-Haenszel fixed-effect meta-analysis.
‡Two trials reported mean blood loss, but one[32] could not be included in effect size estimates because it did not provide SD or CI.
GRADE, Grades of Recommendations, Assessment, Development and Evaluation; IMY, intramyometrial; PPH, postpartum haemorrhage.;

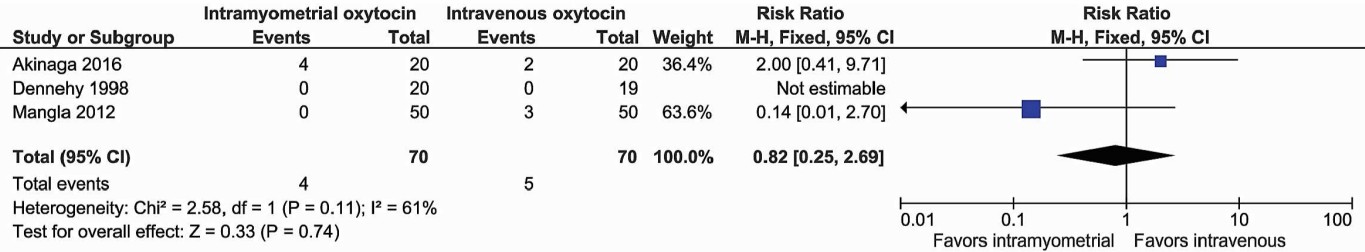

**Figure 3** Forest plot of intramyometrial versus intravenous oxytocin at caesarean. Outcome: need for additional uterotonic. M-H, Mantel-Haenszel.

can also have affected the results because oxytocin plasma levels are dose dependent.[38 39]

It is surprising that we did not find any studies comparing IM versus intravenous prophylactic oxytocin at CS, since these are the more frequently used routes of administration, and there are many trials on this comparison for women giving birth vaginally.[7] While the slower onset of action of oxytocin after IM administration[18 19] can raise concerns about its capacity to control bleeding at CS, this could, in theory, be circumvented by injecting it earlier, for example at uterine incision instead of after fetal delivery. Potential advantages of IM compared with intravenous administration include fewer side effects due to the slower increase in oxytocin plasma concentration, and a longer lasting effect[18 19] which could preclude the need for intravenous oxytocin maintenance infusions. Intravenous oxytocin is a high-alert medication[40] (a drug with increased risk of causing significant patient harm when used in error) which was listed in third place in the list of the top medication errors in 2020.[41] Therefore, in theory, an additional safety advantage of the IM route is that it could potentially reduce medication errors associated with intravenous oxytocin. Finally, for women having a CS, prophylactic IM oxytocin injection would be painless because the patient would be under anaesthesia. It was also surprising to find three trials that assessed IMY oxytocin administration[30–32] because the drug is not licensed for IMY use, and there are no pharmacokinetic (PK) studies on this route of oxytocin administration. In order to licence IMY oxytocin administration, the first step would be to conduct PK/pharmacodynamic (PK/PD) as well as safety studies to ensure that the drug administered in the new route is effective and safe. However, since PK/PD and safety studies are expensive and complex, it may

be difficult to find institutions willing to conduct this type of study.[42]

### Strengths and limitations of the review

Previous reviews have compared various uterotonics (including oxytocin) versus placebo or other agents in different doses, regimens and timing of administration for preventing PPH at CS.[14 43] However, to the best of our knowledge, this is the first systematic review to assess different routes of oxytocin administration for preventing PPH in women giving birth by CS. Strong points of our review include its comprehensive literature search without language restrictions and strict adherence to standard Cochrane methods.[33] Limitations of the review include the small number of trials identified, the lack of response from authors to clarify important methodological and clinical aspects of their studies, and the fact that none of the trials addressed additional core clinical outcomes.

### Implications for practice and research

At present, the limited, low to very low certainty evidence on the effects of IMY versus intravenous oxytocin administration at CS is insufficient to support choosing one route over another for preventing PPH. Our review findings indicate the need for more high-quality trials comparing the effectiveness and safety of different routes of prophylactic oxytocin administration, including the IM route, in women having a CS. Future trials should have adequate sample size, be of high methodological and reporting quality, and recruit participants who represent the general obstetric population, including women at high risk for PPH due to clinical or obstetric disorders, and women undergoing prelabour as well as intrapartum CS, with or without previous exposure to oxytocin for

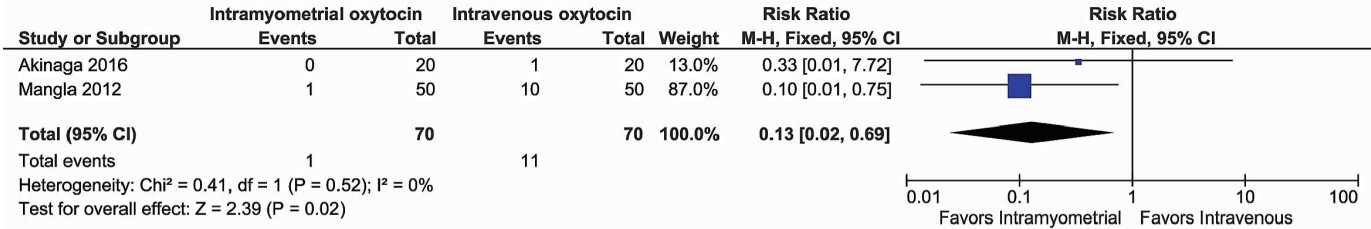

**Figure 4** Forest plot of intramyometrial versus intravenous oxytocin at caesarean. Outcome: nausea and/or vomiting. M-H, Mantel-Haenszel.

labour induction or augmentation. Future studies should consider recent evidence about intravenous dosing regimens to reduce blood loss at CS.[44] These trials also need to report all relevant PPH prevention core outcomes.[35] The results of these future trials are essential for updates of our systematic review and meta-analyses, as well as for individual patient data meta-analyses that will provide more robust evidence on this topic. Given the increasing numbers of births by CS worldwide including in LMIC, and the high rates of maternal mortality following CS in these countries,[45] research to identify the most beneficial route and regimen for prophylactic oxytocin for preventing PPH in women giving birth by CS should be prioritised to avoid preventable mortality and morbidity. Particularly disturbing is the lack of evidence for women having intrapartum CS. These women are particularly vulnerable to uterine atony and increased postpartum bleeding because of prolonged labours, often with previous exposure to oxytocin for induction or augmentation which may desensitise oxytocin receptors and impair myometrial response to prophylactic oxytocin after delivery.[46 47] These evidence gaps are likely to affect more dramatically women in LMIC where skills and resources to provide safe and timely CS, and to deal with its complications, are suboptimal. This lack of evidence is thus likely to increase maternal and perinatal inequalities between and within countries.

## CONCLUSIONS

There is limited, low to very low certainty, evidence on the effects of different routes of prophylactic oxytocin administration at CS for preventing blood loss. The limited evidence suggests that IMY compared with intravenous oxytocin may result in little to no difference in the incidence of PPH and most side effects, but may reduce nausea/vomiting and the volume of blood loss slightly. More trials are needed on this relevant question, especially on the effects of IM oxytocin and involving women with intrapartum CS.

**Author affiliations**
[1]Obstetrics Department, Hospital e Maternidade Santa Joana, Sao Paulo, Brazil
[2]Evidence-Based Healthcare Post-Graduate Program, São Paulo Federal University-UNIFESP, Sao Paulo, Brazil
[3]Anaesthesiology Department, Hospital e Maternidade Santa Joana, Sao Paulo, Brazil
[4]Centre of Health Technology Assessment, Hospital Sirio-Libanes, Sao Paulo, Brazil
[5]Universidade Metropolitana de Santos (UNIMES), Santos, Brazil
[6]Centro Universitário São Camilo, Sao Paulo, Brazil
[7]Reproductive Health and Research, UNDP/UNFPA/UNICEF/WHO/World Bank Special Programme of Research, Development and Research Training in Human Reproduction (HRP), World Health Organization, Geneva, Switzerland

**Contributors** MRT, MS, APB and MW conceptualised the study. MRT wrote the study protocol with the contributions of MS, APB, MW and RR. The search strategy was developed by CdOCL, MRT and RR, with the contribution of MS and APB. Citation screening and study selection were performed by MS, APB, ALCM, RLP and CdOCL under the supervision of RR and MRT. Data extraction and study quality assessment were performed by MRT, ALCM, RR and RLP. Data analysis was conducted by MRT, RR, ALCM and RLP. The report was drafted by MRT with input from MS, APB and MW. All authors reviewed and approved the final manuscript. MRT is the guarantor and takes responsibility for the content of this article.

**Funding** This work was supported by UNDP/UNFPA/UNICEF/WHO/World Bank. Special Programme of Research, Development and Research Training in Human Reproduction (HRP), Department of Sexual and Reproductive Health and Research, WHO, Geneva, Switzerland.

**Competing interests** None declared.

**Patient consent for publication** Not required.

**Provenance and peer review** Not commissioned; externally peer reviewed.

**Data availability statement** All data relevant to the study are included in the article or uploaded as online supplemental information.

**ORCID iDs**
Maria Regina Torloni http://orcid.org/0000-0003-4944-0720
Carolina de Oliveira Cruz Latorraca http://orcid.org/0000-0001-9146-4684

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
