## [Reviewer comments · BMJ Open]

ARTICLE DETAILS

TITLE (PROVISIONAL)	Route of oxytocin administration for preventing blood loss at caesarean section: a systematic review with meta-analysis
AUTHORS	Torloni, Maria Regina; Siaulys, Monica; Riera, Rachel; Martimbianco, Ana; Pacheco, Rafael; Latorraca, Carolina; Widmer, Mariana; Betran, Ana Pilar

VERSION 1 – REVIEW

REVIEWER	Hoaglin, David University of Massachusetts Medical School, Quantitative Health Sciences
REVIEW RETURNED	21-Apr-2021

GENERAL COMMENTS	The authors have thoroughly and thoughtfully reported their systematic review. I find this refreshing. It is unfortunate that their efforts yielded only three trials that compared IMY versus IV administration of oxytocin. The paucity of comparative trials may lie, as they suggest, in the fact that oxytocin is not licensed for IMY use. My comments focus mainly on the statistical aspects of the manuscript, but I suggest that the authors consider commenting (in their Discussion) on the sorts of trials that would be required for licensing and on whether such trials are likely to be conducted. The small number of usable trials and the differences among their designs present a challenge for the analysis. In particular, among the eight outcomes shown in Supplementary file 8, for only one (need for additional uterotonic) did the authors have data from all three trials. Two outcomes had data from two trials, and the other five had data from only one trial. The authors may want to reconsider including “meta-analysis” in the title of their manuscript, because it may promise more than they can deliver. The meta-analyses Beyond the challenges, the meta-analyses have a number of difficulties. Page 7 (of 44), lines 15 and 16 say that “study data were pooled using the software Review Manager 5.4.” It is important to report the actual method of pooling; it is not enough to give only the name of the software. Fortunately (for the purpose of this review) the information on the method is available in the forest plots in Figure 3 and Supplementary file 8: “M-H, Random.” Unfortunately, that method is problematic. Indeed, with little exaggeration, I can say that a Mantel-Haenszel random-effects method does not exist. More charitably, such a method exists only in Review Manager; it has little, if any, support in the meta-analysis literature.
--

The Mantel-Haenszel method (M-H) is a fixed-effect method. In those forest plots, the label “M-H, Random” refers to an option that uses the results of the Mantel-Haenszel method in determining the weights assigned to the individual studies for estimating the between-study variance. That estimated variance is then used, for estimating the pooled effect, in the discredited DerSimonian-Laird random-effects method (DL), which can produce biased estimates with falsely high precision (Cornell et al. 2014). DL may be the default random-effects method in version 5.4 of Review Manager, as it was in version 5.3.

All is not lost, however. The authors have reasonable justification for using a fixed-effect method.

Because sets of studies generally involve some clinical heterogeneity or statistical heterogeneity, the standard advice recommends routinely using a random-effects method for meta-analysis. Accurate estimation of a variance, however, can require quite large samples. When a meta-analysis involves effects from a small number of studies, one cannot expect much accuracy in estimating the between-study variance. Thus, with only three studies it is difficult to justify using a random-effects model.

The forest plots in Figure 3 and Figure 4 indicate that statistical heterogeneity can be neglected. (The value of I^2 is uninformative. It is calculated from Q , on which the chi-square test is based; but that test assumes an incorrect null distribution, and the formula for Q uses that assumption.) Clinical heterogeneity may be another matter; for example, in Table 1 the three studies have rather different dosing schemes for oxytocin (the authors mention this in lines 15 to 17 on page 12). Thus, with appropriate cautions, I suggest using the Mantel-Haenszel fixed-effect method. (In the output from Review Manager in Figure 3, the total numbers of patients are misleading. Because that meta-analysis did not use data from Dennehy 1998, those numbers should be 70 and 70, rather than 90 and 89. The authors, however, could comment on the evidence from Dennehy 1998.)

In Supplementary File 8, Parts A, C (2.3.1), E, F, and G are not meta-analyses. When the data come from only one study, there is no “meta”. The confidence intervals may be useful, but the rest of those forest plots may mislead some readers.

In Part C (2.3.2) of Supplementary File 8, Review Manager correctly reported that the risk ratio is “Not estimable”, because both studies had 0 events in both groups. The authors, however, could still mention that the evidence from those studies is consistent with absence of a difference between IMY and IV.

Table 2: I do not understand the presence of IMY in the Statistical method column. Of course, the table will change if the authors use a fixed-effect analysis. It would be better to use that column for different information: Change the heading to “Effect measure”, and make the entries “Risk Ratio, IMY vs IV” and “Mean Difference, IMY vs IV”. Then add “(95% CI)” to the heading of the Effect size column, and put the meta-analysis method in the footnotes, with a cue (in the Effect size column) on only the entries in the rows numbered 2 and 5. Also, the endpoints of the CI for Mean blood loss are reversed. The lower (in this case, the more negative)

	confidence limit should come first (as in Part G of Supplementary file 8): (-101.71, -13.09). Additional comments Table 1: It is unfortunate that the largest of the three studies, Mangla 2012 (N = 100) has no information on several features. Page 10, line 51: “4/90 versus 5/89” should be “4/70 versus 5/70”. Page 11, line 13: In addition to mean +/- SD for the groups, show the CI for the difference (92.9 - 83.2 = 9.7). CIs are more useful than p-values. Page 11, lines 37 to 40: If Mangla reported a CI for the difference between IMY and IV, it should be possible to derive the standard error of the difference. Figure 1: The selection phase excluded a large number of references (n = 13,383). It would be helpful if that box included a breakdown into several categories. Finally, instead of the results of the analyses, the main message of this manuscript may be the paucity of trials comparing IMY and IV and the need for additional trials (preferably using the same design or closely similar designs, as the result of a consensus among investigators, so that a future meta-analysis can pool their results with fewer limitations; indeed, with appropriate cooperation, such a meta-analysis could be based on individual patient data). Reference Cornell JE, Mulrow CD, Localio R, et al. (2014). Random-effects meta-analysis of inconsistent effects: a time for change. Annals of Internal Medicine 160:267-270.
--	---

REVIEWER	Stjernholm, Ylva Karolinska Institutet, Women´s and Children´s Health
REVIEW RETURNED	23-Apr-2021

GENERAL COMMENTS	The topic route of oxytocin administration at caesarean section is important in clinical practice. Only 3 trials are included in the metaanalysis! However, the manuscript needs some corrections:  1. The authors should address the various cardiotoxic effects of oxytocin - arrhythmia, heart incompensation etc, and water retaining effects - pulmonary oedema etc. more thoroughly, and particularly the disadvantages of a high dose regimen. Studies showing that very small doses of oxytocin < 5 U are sufficient to obtain uterine contraction at cesarean should be mentioned (Charvalho et al etc). 2. The authors should comment the search strategy - search terms, years included etc. 3. The study by Mangla et al 2012 is wrongly cited, since it includes 3 groups - oxytocin IV (n=50), oxytocin IMY before placental detachment (n=50), and oxytocin IMY after placental detachment (n=50). 4. Table 2. is incorrect. More than 1 study reports mean blood loss.
--

	The text after Table 2 states that none of the studies reported blood loss > 100 mL, although this is mentioned in Table 2. Which is right?
--	---

REVIEWER	Stephens, Alexandre Northern NSW Local Health District
REVIEW RETURNED	29-Apr-2021

GENERAL COMMENTS	The authors present the findings of a systematic review and meta-analysis of randomised controlled trials (RCTs) comparing different routes of prophylactic oxytocin administration during caesarean section (CS). This is a well written and presented systematic review and meta-analysis and the authors have comprehensively followed the systematic process required for such studies. I only have a few minor comments below.  1. Study title: I believe "metanalysis" should actually be meta-analysis. 2. Abstract, objectives - for completeness, would it be worth including the primary outcome (blood loss) in the sentence? 3. Page 6, line 36 - objectives - like the abstract, is it worth including the primary study outcome in the sentence? Otherwise, it reads as "To assess the effects of administrating prophylactic oxytocin using different routes in women undergoing a CS", but on what? 4. Page 7, line 38 - it is noted that the certainty of the quality of the evidence was assessed using the GRADE approach (also noted in the abstract). However, this GRADE assessment is not mentioned in the results (I note a reference to Supplementary file 7 in the results but it's not explicit that this refers to the GRADE assessment) or the discussion. Would it be worth including references to / commentary on GRADE in the results and discussion? 5. Page 9, line 52 - I guess it's a very small sample size upon which to calculate I2. Is there a certain limit to the number of studies that are needed before the I2 statistic can be reliably used without too much bias? 6. Page 10, lines 12-13 - please check sentence for missing or incorrect word.
---

VERSION 1 – AUTHOR RESPONSE

Response to Reviewer 1:

Reviewer: Dr. David Hoaglin, University of Massachusetts Medical School
 Comments to the Author: The authors have thoroughly and thoughtfully reported their systematic review. I find this refreshing. It is unfortunate that their efforts yielded only three trials that compared IMY versus IV administration of oxytocin. The paucity of comparative trials may lie, as they suggest, in the fact that oxytocin is not licensed for IMY use. My comments focus mainly on the statistical aspects of the manuscript, but I suggest that the authors consider commenting (in their Discussion) on the sorts of trials that would be required for licensing and on whether such trials are likely to be conducted.

Answer: We have added the following sentence in Discussion: “ In order to license IMY oxytocin administration, the first step would be to conduct pharmacokinetic/pharmacodynamic (PK/PD) as well as safety studies to ensure that the drug administered in the new route is effective and safe. However, since PK/PD and safety studies are expensive and complex, it may be difficult to find institutions willing to conduct this type of study.” We have also added a new reference to support this statement (42).

Reviewer: The small number of usable trials and the differences among their designs present a challenge for the analysis. In particular, among the eight outcomes shown in Supplementary file 8, for only one (need for additional uterotonic) did the authors have data from all three trials. Two outcomes had data from two trials, and the other five had data from only one trial. The authors may want to reconsider including “meta-analysis” in the title of their manuscript, because it may promise more than they can deliver.

Answer: We agree that there are few studies addressing most of our outcomes of interest. If we had not been able to do any meta-analyses at all, it would be misleading to use that word in the title. However, since we did manage to perform more than one meta-analysis, we opted to maintain the word in the title.

Reviewer: Page 7 (of 44), lines 15 and 16 say that “study data were pooled using the software Review Manager 5.4.” It is important to report the actual method of pooling; it is not enough to give only the name of the software.

Answer: We have added the required information to Methods, Data analyses and assessment of the certainty of the evidence (6th sentence): “We pooled data using the Mantel-Haenszel fixed-effect method”.

Reviewer: Because sets of studies generally involve some clinical heterogeneity or statistical heterogeneity, the standard advice recommends routinely using a random-effects method for meta-analysis. Accurate estimation of a variance, however, can require quite large samples. When a meta-analysis involves effects from a small number of studies, one cannot expect much accuracy in estimating the between-study variance. Thus, with only three studies it is difficult to justify using a random-effects model. Thus, with appropriate cautions, I suggest using the Mantel-Haenszel fixed-effect method.

Answer: We agree with the suggestions and we have redone the two meta-analyses (Fig 3 and Fig 4) using the fixed-effects model. We have changed the effect estimates for the meta-analysis on the outcomes PPH prevention and Nausea/Vomiting in the Abstract, Results, Table 2, and Supplementary file 7 (SoF-GRADE) according to the new fixed-effects model.

Reviewer: In the output from Review Manager in Figure 3, the total numbers of patients are misleading. Because that meta-analysis did not use data from Dennehy 1998, those numbers should be 70 and 70, rather than 90 and 89.

Answer: Thank you. We have corrected Figure 3; the totals are now 70 and 70.

Reviewer: In Supplementary File 8, Parts A, C (2.3.1), E, F, and G are not meta-analyses. When the data come from only one study, there is no “meta”. The confidence intervals may be useful, but the rest of those forest plots may mislead some readers.

Answer: Although it is common to present the effect estimate and confidence intervals of single studies in a visual format (forest plot) in Cochrane reviews, we agree that this can mislead some readers. Therefore, we have deleted these figures. Consequently, we have deleted Supplementary file 8. In this revised version of the manuscript, we present all data about the effect size and CI of single studies only in the text of Results and Table 2 of the main manuscript.

Reviewer: In Part C (2.3.2) of Supplementary File 8, Review Manager correctly reported that the risk ratio is “Not estimable”, because both studies had 0 events in both groups. The authors, however, could still mention that the evidence from those studies is consistent with absence of a difference between IMY and IV.

Answer: We have added a new sentence about this to the text of Results, Adverse effects of oxytocin, Hypotension (last sentence): “Although the lack of events did not allow estimation of the risk ratio, the evidence from these trials is consistent with absence of a difference between IMY and IV for this outcome.”

Reviewer: Table 2: I do not understand the presence of IMY in the Statistical method column. Of course, the table will change if the authors use a fixed-effect analysis. It would be better to use that column for different information: Change the heading to “Effect measure”, and make the entries “Risk Ratio, IMY vs IV” and “Mean Difference, IMY vs IV”. Then add “(95% CI)” to the heading of the Effect size column, and put the meta-analysis method in the footnotes, with a cue (in the Effect size column) on only the entries in the rows numbered 2 and 5. Also, the endpoints of the CI for Mean blood loss are reversed. The lower (in this case, the more negative) confidence limit should come first (as in Part G of Supplementary file 8): (-101.71, -13.09).

Answer: We have made all the modifications suggested to Table 2. We corrected the effect estimates for the two meta-analyses according to the fixed-effect analysis.

Additional comments

Reviewer: Table 1: It is unfortunate that the largest of the three studies, Mangla 2012 (N = 100) has no information on several features.

Answer: Yes, it is very unfortunate. We tried to reach this author several times (through emails) to obtain more details about the study but we did not obtain any reply.

Reviewer: Page 10, line 51: “4/90 versus 5/89” should be “4/70 versus 5/70”.

Answer: Thank you. We have corrected the denominators.

Reviewer: Page 11, line 13: In addition to mean +/- SD for the groups, show the CI for the difference (92.9 - 83.2 = 9.7). CIs are more useful than p-values.

Answer: We have added this information to this sentence.

Reviewer: Page 11, lines 37 to 40: If Mangla reported a CI for the difference between IMY and IV, it should be possible to derive the standard error of the difference.

Answer: Unfortunately Mangla et al only reported the mean blood loss in the groups (no standard deviation and no CI). The authors did not reply to our emails asking for this information and additional details.

Reviewer: Figure 1: The selection phase excluded a large number of references (n = 13,383). It would be helpful if that box included a breakdown into several categories.

Answer: In standard (including Cochrane) systematic reviews, it is not expected that the investigators record and report the reasons for excluding studies in the screening phase (reading of titles and abstracts of all studies identified through the search)^a. Therefore, we did not collect and cannot provide this information. The large number of references retrieved was probably due to our comprehensive search, without language restrictions, in seven electronic databases, two trial registries, and one grey literature base. This comprehensive type search, with a relatively low specificity, is also common in systematic reviews; it is used to minimize the chances of missing any relevant studies.

a. Lefebvre C, Glanville J, Briscoe S et al. Searching for and selecting studies. In: Higgins JPT, Thomas J, Chandler J, Cumpston M, Li T, Page MJ, Welch VA (editors). Cochrane Handbook for Systematic Reviews of Interventions version 6.2 (updated February 2021). Cochrane, 2021)

Reviewer: Finally, instead of the results of the analyses, the main message of this manuscript may be the paucity of trials comparing IMY and IV and the need for additional trials (preferably using the same design or closely similar designs, as the result of a consensus among investigators, so that a future meta-analysis can pool their results with fewer limitations; indeed, with appropriate cooperation, such a meta-analysis could be based on individual patient data).

Answer: We agree that the main message of the review should be the paucity of trials comparing different routes of administration of oxytocin (not only IMY x IV) to prevent post-partum bleeding in women having a CS, and therefore, the need for more trials to assess the optimal route. We state these messages in two sentences of our Conclusion: “There is limited, low to very low certainty evidence on the effects of different routes of prophylactic oxytocin administration at CS.” (1st sentence of Conclusion), and “More trials are needed on this relevant question, especially on the effects of intramuscular oxytocin and involving women with intrapartum CS.” (last sentence of Conclusion). We also provide suggestions of important aspects of these future trials in the section on Implications for Practice and Research of our Discussion (2nd, 3rd and 4th sentences). We have added a sentence in Discussion about the inclusion of these new results, including individual patient data, in future updates

of this systematic review and meta-analysis (5th sentence of Discussion/ Implications for Practice and Research): “The results of these future trials are essential for updates of our systematic review and meta-analyses, as well as for individual patient data metanalyses that will provide more robust evidence on this topic”.

Response to Reviewer 2:

Dr. Ylva Stjernholm, Karolinska Institutet Comments to the Author: The topic route of oxytocin administration at caesarean section is important in clinical practice. Only 3 trials are included in the metaanalysis! However, the manuscript needs some corrections:

1. The authors should address the various cardiotoxic effects of oxytocin - arrhythmia, heart incompensation etc, and water retaining effects - pulmonary oedema etc. more thoroughly, and particularly the disadvantages of a high dose regimen. Studies showing that very small doses of oxytocin < 5 U are sufficient to obtain uterine contraction at cesarean should be mentioned (Charvalho et al etc).

Answer: These are important adverse effects of oxytocin. We have five sentences and 12 references (13,16 and17-26) in the third paragraph of Introduction describing the cardiovascular effects of oxytocin in women having a CS. Due to total word count limitation, we cannot expand more on this topic but we believe it is clearly and directly covered.

According to a recently published systematic review (Phung et al, Am J Obstet Gynecol. 2021 May 3:S0002-9378(21)00541-X. Epub ahead of print. PMID: 33957113), there is limited evidence on the ideal IV dosing regimen of oxytocin to prevent blood loss at CS. Since our systematic review aimed to assess the best route (not the dose) of oxytocin administration at CS, we focused our manuscript on this topic. However, we added a sentence (and this new reference) to Discussion/Implications for Practice and Research to refer to this important issue: “Future studies should consider recent evidence about IV dosing regimens to reduce blood loss at CS.⁴⁴”

2. The authors should comment the search strategy - search terms, years included etc.

Answer: We have added the main search terms to Methods, Search strategy section (1st sentence): “We created a search strategy with the following general terms and synonyms: “caesarean section” or “C-section” or “abdominal delivery” and “oxytocin” (details in Supplementary file 2).” The last sentence of this section was modified as follows: “Searches were run from database inception until May 24, 2020.”

3. The study by Mangla et al 2012 is wrongly cited, since it includes 3 groups - oxytocin IV (n=50), oxytocin IMY before placental detachment (n=50), and oxytocin IMY after placental detachment (n=50).

Answer: As we described in Supplementary file 4-Study details, the Mangla 2012 trial had three arms. However, we included in the review only the two arms (Groups 1 and 2) that compared different routes of oxytocin administration (IV x IMY) given at the same time (i.e. after placental separation), since the objective of this systematic review was to compare different routes (and not timings) of administration. To clarify this in the text of Results, we have modified the sentence that describes this study (2nd paragraph): “Mangla et al randomised 150 women in India to start a 500 mL infusion containing oxytocin after placental separation (Group 1, N=50), or to receive an IMY oxytocin bolus injection (half in each uterine cornus) either after placental separation (Group 2, N=50) or before placental separation (Group 3, N=50)³². We did not include in this systematic review the participants in the third group because it compared a different route and a different timing of oxytocin administration (Supplementary file 4)”. We have also added a new footnote to Table 1 explaining the sample size of this trial.

4. Table 2 is incorrect. More than 1 study reports mean blood loss.

Answer: Indeed, as presented in Table 1, two trials (Mangla 2012 and Akinaga 2016) reported mean blood loss. However, Mangla 2012 could not be included in Table 2 that presents the effect estimates because this trial did not provide the standard deviation or confidence interval around the mean volume of blood loss. We had reported this in the text of Results, Volume of blood loss (1st sentence). However,

to better clarify this fact, we added a new footnote to Table 2: “2. Two trials reported mean blood loss, but one (Mangla 2012) could not be included in effect size estimates because it did not provide standard deviation or CI.”

Response to Reviewer 3:

Dr. Alexandre Stephens, Northern NSW Local Health District Comments to the Author: The authors present the findings of a systematic review and meta-analysis of randomised controlled trials (RCTs) comparing different routes of prophylactic oxytocin administration during caesarean section (CS). This is a well written and presented systematic review and meta-analysis and the authors have comprehensively followed the systematic process required for such studies. I only have a few minor comments below.

1. Study title: I believe "metanalysis" should actually be meta-analysis.

Answer: We changed the spelling of the word throughout the manuscript.

2. Abstract, objectives - for completeness, would it be worth including the primary outcome (blood loss) in the sentence?

Answer: As suggested by the reviewer we have changed the sentence in Abstract, Objectives to: “Assess the effects of different routes of prophylactic oxytocin administration to prevent blood loss at caesarean section (CS)”, as stated in the title.

3. Page 6, line 36 - objectives - like the abstract, is it worth including the primary study outcome in the sentence? Otherwise, it reads as "To assess the effects of administering prophylactic oxytocin using different routes in women undergoing a CS", but on what?

Answer: As suggested by the reviewer, we have changed the sentence in Objectives (in the manuscript) to: “To assess the effects of different routes of prophylactic oxytocin administration to prevent blood loss at CS”, as stated in the title.

4. Page 7, line 38 - it is noted that the certainty of the quality of the evidence was assessed using the GRADE approach (also noted in the abstract). However, this GRADE assessment is not mentioned in the results (I note a reference to Supplementary file 7 in the results but it's not explicit that this refers to the GRADE assessment) or the discussion. Would it be worth including references to / commentary on GRADE in the results and discussion?

Answer: To address the reviewer's comment, in the Results section, we have changed the sentence that refers to Supplementary file 7 as follows: “See supplementary file 7 for the assessment of the certainty of the evidence (GRADE) for each outcome.” Also in the Results section, we report the certainty of the evidence for each outcome, after the effect estimates, as the last information provided in the parentheses. For example, for Incidence of PPH: “(0/50 versus 3/50, RR 0.14, 95% CI 0.01 to 2.70; 100 participants; 1 RCT, low certainty evidence). We have added a new column to Table 2 with the certainty of the evidence for each comparison.

In the Discussion section, we commented that the certainty of the evidence is low to very low (last sentence of 1st paragraph): “The overall certainty of the evidence was low or very low for all outcomes due mainly to imprecision.” To clarify the meaning of this statement to readers not familiar with this terminology, we have added one more sentence: “According to the GRADE approach, this means that we have limited confidence in the effect estimates, and that the true effect may be substantially different from the estimates presented³⁷.” We also allude to the certainty of the evidence in Conclusions (1st sentence): “There is limited, low to very low certainty, evidence on the effects of different routes of prophylactic oxytocin administration at CS.”

5. Page 9, line 52 - I guess it's a very small sample size upon which to calculate I2. Is there a certain limit to the number of studies that are needed before the I2 statistic can be reliably used without too much bias?

Answer: In 2002, Higgins and Thomas introduced the I^2 statistics to improve the classical test (Cochran's Q) used to assess heterogeneity in meta-analyses. However, like Cochran's Q, the I^2 statistic can also be influenced by the number of studies included in the meta-analysis. Although small numbers of studies tend to increase bias, there is no specific number of studies that must be included in a meta-analysis before the I^2 statistic can be considered unbiased. In many small meta-analyses, we may not be able to estimate heterogeneity with much precision. No statistic can change the limitations of small meta-analyses (von Hippel PT. BMC Med Res Methodol. 2015 Apr 14;15:35).

6. Page 10, lines 12-13 - please check sentence for missing or incorrect word.

Answer: Thank you. We have corrected the sentence to "The maximum HR in the IV group was significantly higher than in the IMY group".

VERSION 2 – REVIEW

REVIEWER	Hoaglin, David University of Massachusetts Medical School, Quantitative Health Sciences
REVIEW RETURNED	28-Jun-2021

GENERAL COMMENTS	I thank the authors for their careful responses to my comments on the previous version. I am glad that my suggestions were helpful. I have only a few additional comments. A comment from Reviewer 3 drew my attention to I^2, on which I did not comment previously. I generally advise authors to avoid using I^2 (and the Q-test for heterogeneity). Thus, I suggest that the authors omit I^2 in lines 51 to 53 on page 7 and in line 18 on page 13. The usual test based on Q is not valid, because it uses an incorrect null distribution (Hoaglin 2016). The same problem undermines the I^2 measure, which is usually calculated from Q. Those measures face a further complication because the true null distribution of Q differs among outcome measures, and it generally involves parameters that must be estimated from the data in the meta-analysis. Page 4, line 22ff: The authors cite a systematic review in 2020 (Reference 7). Out of curiosity, I looked at that systematic review. I was disappointed to see that the text gives little information on the methods used for the meta-analyses. I was also disappointed, but not surprised, to see (page 37ff) "Risk Ratio (IV, Random, 95% CI)." Having read the technical documentation for methods of meta-analysis in the Cochrane Collaboration's Review Manager, I recognized "IV, Random" as shorthand for the DerSimonian-Laird random-effects method ("IV" refers to its use of inverse-variance weights). Despite its great popularity, that method is unreliable. As I mentioned in a comment on the previous version, it can produce biased estimates with falsely high precision (Cornell et al. 2014). Also, its confidence intervals have below-nominal coverage and are inferior to those produced by another method (IntHout et al. 2014). As the dates of those publications suggest, the shortcomings of the DerSimonian-Laird method have been discussed in the meta-analysis literature for many years. Thus, it is discouraging to find that method in a systematic review published in 2020. In fairness, it would be awkward if the authors of a
--

	Cochrane review declined to use a method endorsed in the Cochrane Collaboration's own software. More importantly, it is unfortunate that (page 4, line 26) "Based on these findings, WHO updated its guideline in 2020". WHO should revisit that decision. In the present manuscript, the authors could acknowledge the difficulties in the Cochrane review by changing "showed" to "estimated" in line 21, and by inserting in line 29 (preceding "For women giving birth by CS") something like the following: "This guideline, however, should be interpreted with caution. A reviewer pointed out that the meta-analyses in the systematic review[Reference 7] used an unreliable method." I am glad to turn now to minor, less-technical comments. Page 2, line 14: "metanalyses" should be "meta-analyses". This error occurs also in line 45 on page 7 and in line 26 on page 16. I wonder whether the authors are the victims of some type of autocorrect. Page 2, lines 34 and 36 (also page 16, line 8): "At present, there is insufficient evidence to recommend IMY or IV oxytocin administration at CS to prevent blood loss." Some, perhaps many, readers may interpret this sentence as saying that the evidence does not support recommending IMY and that the evidence also does not support recommending IV. I think it is not what the authors intend to say. All their comparisons are between IMY and IV. So they could say something like "The evidence does not support choosing one of IMY and IV instead of the other." Compare this statement with paragraph 2 of the Introduction. Page 7, line 50: Can "systematic mean differences (SMD)" be omitted? I did not see results that used SMD. References Cornell JE, Mulrow CD, Localio R, et al. (2014). Random-effects meta-analysis of inconsistent effects: a time for change. Annals of Internal Medicine 160:267-270. Hoaglin DC (2016). Misunderstandings about Q and 'Cochran's Q Test' in meta-analysis. Statistics in Medicine 35:485-495. IntHout J, Ioannidis JPA, Borm GF (2014). The Hartung-Knapp-Sidik-Jonkman method for random effects meta-analysis is straightforward and considerably outperforms the standard DerSimonian-Laird method. BMC Medical Research Methodology 14:25.
--	---

VERSION 2 – AUTHOR RESPONSE

Reviewer: 1

Dr. David Hoaglin, University of Massachusetts Medical School

Comments to the Author:

I thank the authors for their careful responses to my comments on the previous version. I am glad that my suggestions were helpful. I have only a few additional comments. A comment from Reviewer 3 drew my attention to I^2 , on which I did not comment previously. I generally advise authors to avoid using I^2 (and the Q-test for heterogeneity). Thus, I suggest that the authors omit I^2 in lines 51 to 53 on page 7 and in line 18 on page 13. The usual test based on Q is not valid, because it uses an incorrect null distribution (Hoaglin 2016). The same problem undermines the I^2 measure, which is usually calculated from Q. Those measures face a further complication because the true null distribution of Q differs among outcome measures, and it generally involves parameters that must be estimated from the data in the meta-analysis.

Response: We acknowledge that there are limitations in the measures of inconsistency in meta-analyses. However, with all due respect, we did not follow the suggestion to omit reporting I^2 in our article. Reporting measures of heterogeneity in meta-analyses is part of the standard methods, and a mandatory item, in Cochrane reviews (Higgins JPT, Thomas J, Chandler J, et al. Cochrane Handbook for Systematic Reviews of Interventions version 6.1, 10.10.2). Moreover, reporting of heterogeneity in meta-analyses is also a required field in the PRISMA reporting guideline (Checklist Item 14 in Methods, and 21 in Results). Finally, inconsistency is one of the five elements used in the GRADE approach to assess the certainty (quality) of the body of evidence for each outcome.

Page 4, line 22ff: The authors cite a systematic review in 2020 (Reference 7). Out of curiosity, I looked at that systematic review. I was disappointed to see that the text gives little information on the methods used for the meta-analyses. I was also disappointed, but not surprised, to see (page 37ff) “Risk Ratio (IV, Random, 95% CI).” Having read the technical documentation for methods of meta-analysis in the Cochrane Collaboration’s Review Manager, I recognized “IV, Random” as shorthand for the DerSimonian-Laird random-effects method (“IV” refers to its use of inverse-variance weights). Despite its great popularity, that method is unreliable. As I mentioned in a comment on the previous version, it can produce biased estimates with falsely high precision (Cornell et al. 2014). Also, its confidence intervals have below-nominal coverage and are inferior to those produced by another method (IntHout et al. 2014). As the dates of those publications suggest, the shortcomings of the DerSimonian-Laird method have been discussed in the meta-analysis literature for many years. Thus, it is discouraging to find that method in a systematic review

published in 2020. In fairness, it would be awkward if the authors of a Cochrane review declined to use a method endorsed in the Cochrane Collaboration's own software. More importantly, it is unfortunate that (page 4, line 26) "Based on these findings, WHO updated its guideline in 2020". WHO should revisit that decision. In the present manuscript, the authors could acknowledge the difficulties in the Cochrane review by changing "showed" to "estimated" in line 21, and by inserting in line 29 (preceding "For women giving birth by CS") something like the following: "This guideline, however, should be interpreted with caution. A reviewer pointed out that the meta-analyses in the systematic review [Reference 7] used an unreliable method."

Response: We have made the change suggested in line 21. We did not insert the sentence suggested in line 29. As the reviewer stated, Cochrane expects that authors follow standard, recommended, methods in all its reviews, and the World Health Organization uses Cochrane reviews to support its recommendations. We understand that challenging the methods for meta-analysis used by the Cochrane Collaboration (or any state-of-the-art method used in any discipline) is important to advance knowledge and techniques. However, we respectfully suggest that the technical discussion and questioning of the appropriateness of the methods for meta-analysis proposed by the Cochrane Collaboration is beyond the scope of our manuscript. We think that this topic deserves a separate debate and could be addressed in another manuscript or methodological commentary paper. Meanwhile, the reviewer's comments and author's response will be available to all readers since BMJ Open publishes this material with the article.

I am glad to turn now to minor, less-technical comments.

Page 2, line 14: "metanalyses" should be "meta-analyses". This error occurs also in line 45 on page 7 and in line 26 on page 16. I wonder whether the authors are the victims of some type of autocorrect.

Response: We have corrected the errors mentioned.

Page 2, lines 34 and 36 (also page 16, line 8): "At present, there is insufficient evidence to recommend IMY or IV oxytocin administration at CS to prevent blood loss." Some, perhaps many, readers may interpret this sentence as saying that the evidence does not support recommending IMY and that the evidence also does not support recommending IV. I think it is not what the authors intend to say. All their comparisons are between IMY and IV. So they could say something like "The evidence does not support choosing one of IMY and IV

instead of the other.” Compare this statement with paragraph 2 of the Introduction.

Response: We have modified the sentence in both sections (Abstract and Discussion) to clarify the meaning of our findings.

Page 7, line 50: Can “systematic mean differences (SMD)” be omitted? I did not see results that used SMD.

Response: We have eliminated this part of the sentence, as suggested.